# A pan-cancer analysis of PBAF complex mutations and their association with immunotherapy response

A. Ari Hakimi [1,2✉], Kyrollis Attalla[1], Renzo G. DiNatale[1], Irina Ostrovnaya[3], Jessica Flynn[3], Kyle A. Blum[1], Yasser Ged[4], Douglas Hoen[2], Sviatoslav M. Kendall[3,5], Ed Reznik [6], Anita Bowman[7], Jason Hwee[7], Christopher J. Fong[5,8], Fengshen Kuo [2], Martin H. Voss [4], Timothy A. Chan [2,5,9,10,11,12] & Robert J. Motzer [4]

There is conflicting data regarding the role of PBAF complex mutations and response to immune checkpoint blockade (ICB) therapy in clear cell renal cell carcinoma (ccRCC) and other solid tumors. We assess the prevalence of PBAF complex mutations from two large cohorts including the pan-cancer TCGA project ($n = 10,359$) and the MSK-IMPACT pan-cancer immunotherapy cohort ($n = 3700$). Across both cohorts, PBAF complex mutations, predominantly *PBRM1* mutations, are most common in ccRCC. In multivariate models of ccRCC patients treated with ICB ($n = 189$), loss-of-function (LOF) mutations in *PBRM1* are not associated with overall survival (OS) (HR = 1.24, $p = 0.47$) or time to treatment failure (HR = 0.85, $p = 0.44$). In a series of 11 solid tumors ($n = 2936$), LOF mutations are not associated with improved OS in a stratified multivariate model (HR = 0.9, $p = 0.7$). In a current series of solid tumors treated with ICB, we are unable to demonstrate favorable response to ICB in patients with PBAF complex mutations.

[1] Urology Service, Department of Surgery, Memorial Sloan Kettering Cancer Center, New York, NY, USA. [2] Immunogenomics and Precision Oncology Platform, Memorial Sloan Kettering Cancer Center, New York, NY, USA. [3] Department of Epidemiology and Biostatistics, Memorial Sloan Kettering Cancer Center, New York, NY, USA. [4] Department of Medicine, Memorial Sloan Kettering Cancer Center, New York, NY, USA. [5] Human Oncology and Pathogenesis Program, Memorial Sloan Kettering Cancer Center, New York, NY, USA. [6] Computational Oncology Service, Department of Epidemiology and Biostatistics, Memorial Sloan Kettering Cancer Center, New York, NY, USA. [7] Department of Pathology, Memorial Sloan Kettering Cancer Center, New York, NY, USA. [8] Marie-Josée and Henry R. Kravis Center for Molecular Oncology, Memorial Sloan Kettering Cancer Center, New York, NY, USA. [9] Department of Radiation Oncology, Memorial Sloan Kettering Cancer Center, New York, NY, USA. [10] Center for Immunotherapy and Precision Immuno-Oncology, Cleveland Clinic, Cleveland, OH, USA. [11] Lerner Research Institute, Cleveland Clinic, Cleveland, OH, USA. [12] Taussig Cancer Center, Cleveland Clinic Case Comprehensive Cancer Center, Cleveland, OH, USA. ✉email: hakimia@mskcc.org

mmune-checkpoint blockade (ICB) therapy has revolutionized the treatment of many malignancies, leading to an extensive search for predictive and prognostic biomarkers. We and others have reported on the association of ICB response with tumor mutation burden (TMB), neoantigen load and clonality, copy number alterations (CNA), microsatellite instability, and human leukocyte antigen zygosity across a variety of cancer types[1–3]. Additional evidence has linked ICB response to the tumor microenvironment (TME), specifically T cell inflammation[4], and integrated analyses have looked at both TMB and T cell inflammation in joint models[5]. Despite these consistent signals across different studies and cancer types, there is a substantial number of tumors with lower mutation and neoantigen burdens that respond to ICB. Indeed, even studies that link response to TMB or neoantigen burden often have overlap between responders and non-responders.

Recent work from several groups pointed to the association of ICB response and mutations in the SWI/SNF chromatin remodeling complex, more specifically the polybromo and BRG-1 associated factors (PBAF) complex which includes the genes *ARID2*, *PBRM1*, and *BRD7* (refs. [6–8]). Inactivation of the gene encoded by the PBAF complex was recently found to sensitize melanoma cells to T cell-specific killing[7]. Miao et al.[6] demonstrated that in a series of nearly 100 metastatic clear cell renal cell carcinoma (ccRCC) patients, those harboring loss-of-function (LOF) mutations in *PBRM1* had clinical benefit from ICB. They further demonstrated that in other microsatellite stable tumors such as melanoma, lung, bladder, and head and neck cancers, loss of PBAF was also associated with clinical response[8]. Similarly, a recent report validated the association between *PBRM1* alterations and ICB response[9] in CheckMate 025, a randomized phase 3 trial of nivolumab versus everolimus which demonstrated a survival benefit for nivolumab in the second- and third-line setting[10]. Further functional and transcriptomic analysis suggested that *PBRM1*-deficient tumors possessed altered immune signaling pathways.

However, in a recent randomized phase II study of metastatic ccRCC, no association was seen between presence of *PBRM1* mutations and treatment response to the PD-L1-directed atezolizumab, nor to the combination of atezolizumab plus bevacizumab ($n = 136$); there was a favorable effect on treatment response in patients receiving sunitinib (anti-VEGF) on the control arm of the same study ($n = 72$)[4]. Given the discordant clinical data in PBAF complex loss, as well as its potential impact on the TME, we seek to leverage several large clinical trial data sets with genomic data along with our inhouse ICB-treated patients to explore the effects of PBAF loss on the TME and clinical outcomes. We utilize two large pan-cancer cohorts to determine the frequency of PBAF mutations, and we explore the prognostic significance of PBAF mutations across various solid-tumor malignancies in our Memorial Sloan Kettering Cancer Center (MSKCC) ICB cohort. Finally, we assess the impact of *PBRM1* LOF mutations on TME expression programs using a cohort of 594 ccRCC patients with transcriptomic data. We are ultimately unable to demonstrate a favorable response to ICB in patients with PBAF complex mutations and further, gene-expression analysis of *PBRM1* mutated metastatic ccRCC patients demonstrate consistent upregulation in hypoxia inducible factor (HIF) signaling and angiogenesis, but inconsistent interferon gamma signaling and other immune response pathways.

## Results
**Presence of PBAF mutations in TCGA across cancers**. To evaluate the prevalence of PBAF complex mutations, we queried the pan-cancer TCGA atlas ($n = 10,359$) and analyzed all three genes in the complex (*PBRM1*, *ARID2*, and *BRD7*). Overall, 7.7% of all tumors possessed any PBAF complex mutation; incidence among malignancies included in the pan-cancer TCGA cohort was highest in ccRCC (KIRC) particularly for *PBRM1* mutations, followed by melanoma (SKCM), cholangiocarcinoma (CHOL), stomach (STAD), uterine (UCEC), and bladder (BLCA) cancers (Fig. 1a, b). Additionally, highly mutated tumors were more likely to possess PBAF complex mutations (Fig. 2). Mutations in *PBRM1*, *ARID2*, and *BRD7* each represented about 3.8% (52.7% LOF), 3.6% (39% LOF), and 1.1% (34.4% LOF), respectively (Supplementary Fig. 1).

**MSK-IMPACT immunotherapy pan-cancer cohort**. We first assessed the incidence of *PBRM1* and *ARID2* mutations (*BRD7* not included in IMPACT) among all patients treated with ICB ($n = 3700$). For the MSK-IMPACT cohort, we restricted our analysis to 189 ccRCC patients and 2936 patients treated with immunotherapy comprising 11 other cancer types that had a minimum of 50 patients and 5 *PBRM1* or *ARID2* mutants. Clinical characteristics of the included cohort, including age, gender and drug class, as well as TMB and fraction genome altered (FGA) vary substantially across cancer types and are included in Table 1. Available PBAF complex mutations included *PBRM1* and *ARID2*, present at 7.4% and 6.5%, respectively, across the pan-cancer cohort; LOF frequencies are 3.9% and 2.3%, respectively.

Consistent with the TCGA analysis, *PBRM1* mutations were most common in ccRCC patients (46.6%), followed by non-melanoma skin cancer (9%) and melanoma (8%), while *ARID2* was most common in melanoma (13%), followed by non-melanoma skin cancer and colorectal cancer (11%) (Supplementary Fig. 2). With the exception of ccRCC, several of the tumors harboring PBAF complex mutations were highly mutated cancer types (Fig. 3a, b). This included both LOF mutations (frameshift and nonsense mutations) as well as missense mutations.

**PBAF mutation and response to immunotherapy in ccRCC**. Given the findings of Miao et al.[6] and Braun et al.[9] with respect to *PBRM1* LOF mutations and response to ICB in RCC, we analyzed our cohort of ICB-treated metastatic ccRCC patients ($n = 189$) with more detailed clinical annotations including International Metastatic Renal Cell Carcinoma Database Consortium (IMDC) prognostic score ($n = 180$; Table 2), treatment details and outcomes with therapy, including time-to-treatment failure (TTF). *PBRM1* LOF mutations were present in 61 of these 189 patients (32%), and non-LOF mutations were found in 27 patients (14%). Since *ARID2* mutations were only present in six patients (4 of them LOF) and might have a distinct effect on outcome, we further analyzed only *PBRM1* mutations. There were 57 deaths and 147 treatment failures among these 189 patients, with a median overall survival (OS) of 68.2 months (95% CI 44.4, NA) and median TTF of 8.9 months (95% CI 6.9, 12.42), but no difference for either outcome when comparing patients with *PBRM1* LOF to others (Fig. 4). *PBRM1* mutation rate was not significantly different in patients who received first line ($n = 97$, 30% LOF mutations) or second or higher line ($n = 92$, 35%) ICB or ICB/VEGF combinations. *PBRM1* mutations were not associated with TTF in the entire ccRCC cohort (LOF HR 0.73, $p = 0.11$; non-LOF HR 1.05, $p = 0.84$) and not significantly associated with OS (LOF HR = 1.5, $p = 0.16$; non-LOF HR = 1.05, $p = 0.91$) (Table 3). When comparing outcomes with first-line ICB therapy in patients with *PBRM1* LOF mutations vs. wild type, no significant differences were seen for TTF (HR = 0.6, $p = 0.075$) or OS (HR = 1.7, $p = 0.29$); similarly, no differences were seen for those receiving ICB in the second line or higher (TTF HR = 0.87,

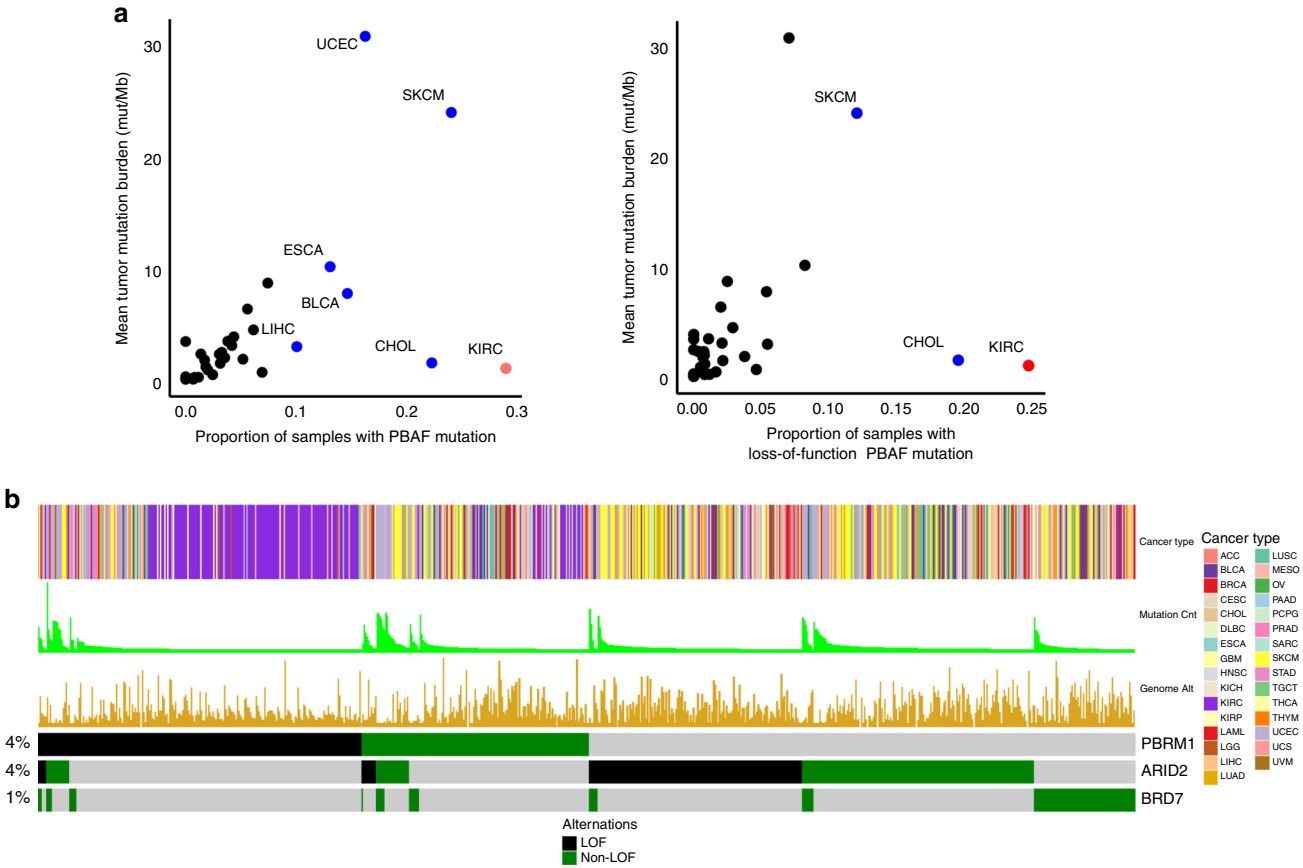

**Fig. 1 PBAF complex mutations across The Cancer Genome Atlas (TCGA) *n* = 10,359. a** All PBAF complex mutations as a function of mean tumor burden (left) and loss of function (LOF) only mutations (right). **b** OncoPrint plot demonstrating loss-of-function vs. non-LOF PBAF complex mutations across the TCGA.

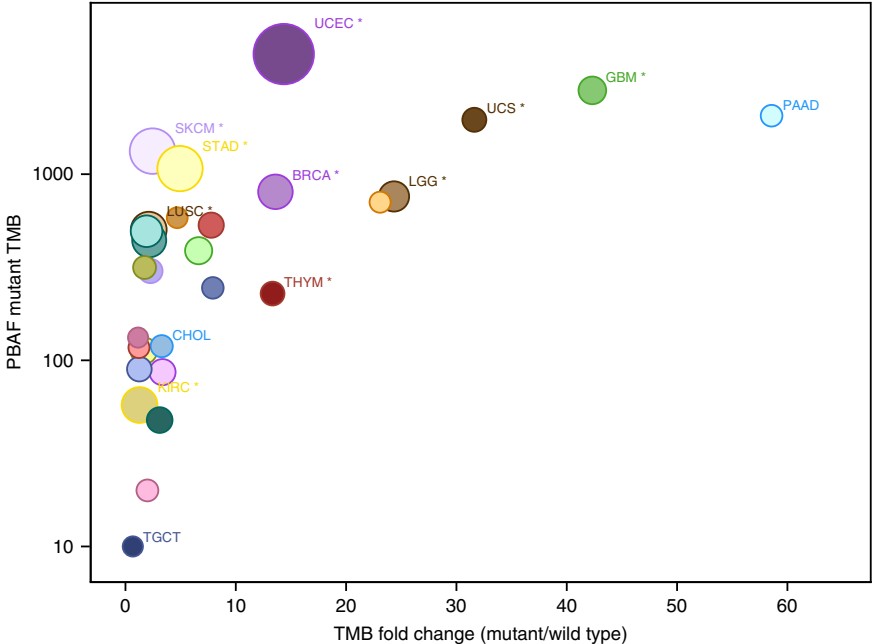

**Fig. 2 Mutation burden in PBAF complex altered tumors.** Tumor mutation burden (TMB) of PBAF complex mutated tumors in the TCGA plotted against TMB ratio of mutated tumors vs. wild type.

**Table 1 Patient characteristics by cancer type in MSK IMPACT cohort.**

| Variable | Age | Gender | | Tumor mutation burden score | Fraction genome altered (median) | Drugs | | |
| | | F | M | | | CTLA-4 | CTLA-4 \| PD-1/PD-L1 | PD-1/PD-L1 |
|---|---|---|---|---|---|---|---|---|
| Overall ($n = 3125$) | 64 (15, 90) | 1310 (41.9) | 1815 (58.1) | 6.1 (0, 368.6) | 0.48 | 24 (1) | 681 (21.8) | 2420 (77.4) |
| Bladder cancer ($n = 249$) | 69 (32, 90) | 61 (24.5) | 188 (75.5) | 7.9 (0, 209.5) | 0.56 | 0 (0) | 45 (18.1) | 204 (81.9) |
| Cancer of unknown primary ($n = 117$) | 64 (17, 89) | 54 (46.2) | 63 (53.8) | 5.3 (0, 90.4) | 0.51 | 1 (0.9) | 15 (12.8) | 101 (86.3) |
| Colorectal cancer ($n = 145$) | 56 (19, 88) | 60 (41.4) | 85 (58.6) | 8.8 (0, 368.6) | 0.37 | 1 (0.7) | 13 (9) | 131 (90.3) |
| Endometrial cancer ($n = 129$) | 66 (40, 90) | 129 (100) | 0 (0) | 6.1 (0, 156.4) | 0.43 | 0 (0) | 19 (14.7) | 110 (85.3) |
| Esophagogastric cancer ($n = 173$) | 62 (23, 87) | 40 (23.1) | 133 (76.9) | 4.9 (0, 62) | 0.62 | 1 (0.6) | 45 (26) | 127 (73.4) |
| Glioma ($n = 177$) | 54 (15, 82) | 67 (37.9) | 110 (62.1) | 4.4 (0, 330) | 0.29 | 0 (0) | 4 (2.3) | 173 (97.7) |
| Head and neck cancer ($n = 164$) | 62 (17, 84) | 36 (22) | 128 (78) | 5.3 (0, 68.5) | 0.43 | 0 (0) | 11 (6.7) | 153 (93.3) |
| Hepatobiliary cancer ($n = 81$) | 65 (15, 87) | 37 (45.7) | 44 (54.3) | 3.5 (0, 50.9) | 0.38 | 1 (1.2) | 3 (3.7) | 77 (95.1) |
| Melanoma ($n = 607$) | 66 (16, 90) | 219 (36.1) | 388 (63.9) | 9.7 (0, 181.8) | 0.48 | 18 (3) | 361 (59.5) | 228 (37.6) |
| Non-small cell lung cancer ($n = 1041$) | 67 (23, 90) | 547 (52.5) | 494 (47.5) | 7 (0, 100.4) | 0.58 | 1 (0.1) | 123 (11.8) | 917 (88.1) |
| Clear cell renal cell carcinoma ($n = 189$) | 61 (35, 84) | 47 (25) | 142 (75) | 3.9 (0, 22.6) | 0.36 | 0 (0) | 35 (18.5) | 154 (81.5) |
| Skin cancer, non-melanoma ($n = 53$) | 70 (35, 90) | 13 (24.5) | 40 (75.5) | 2 (0, 179.1) | 0.29 | 1 (1.9) | 7 (13.2) | 45 (84.9) |

Median (range) reported for continuous variables and % for categorical variables.

$p = 0.61$; OS HR = 1.71.3, $p = 0.44$) (Fig. 5). In a multivariate model adjusted for TMB and drug class (significant predictors of progression free survival), PBRM1 was not significantly associated with TTF (LOF HR = 0.85, 95% CI 0.57, 1.28, $p = 0.44$; non-LOF HR = 1.22, 95% CI 0.77, 1.94, $p = 0.4$). Similarly, in the model for OS adjusted for IMDC risk score and line of therapy (significant predictors of OS), PBRM1 was not significant (LOF HR = 1.24, 95% CI 0.69, 2.25, $p = 0.47$; non-LOF HR = 0.88, 95% CI 0.36, 2.14, $p = 0.78$).

**PBAF complex mutation and ICB outcomes in other cancer types.** To assess the impact of PBAF complex mutations in non-RCC cohorts treated with ICB profiled with MSK-IMPACT, we restricted our analysis to 11 tumor types with at least 50 patients and at least 5 patients with PBRM1 or ARID2 mutations ($n = 2936$). These included bladder, colorectal, non-small-cell lung, esophagogastric, endometrial, non-melanoma skin, hepatobiliary, head and neck cancers, melanoma, glioma, and cancer of unknown primary. Overall prevalence was 4.9% for PBRM1 (2% of them LOF) and 6.7% for ARID2 (3% of them LOF). PBRM1 mutations were not significantly associated with OS in a cohort of 11 cancer types in a Cox model stratified by cancer type (LOF HR = 0.9, 95% CI 0.6,1.4, $p = 0.7$; non-LOF HR 1.03, 95% CI 0.73,1.5, $p = 0.86$), and remained insignificant after adjusting for TMB and total CNA (LOF HR = 1.2, 95% CI 0.8,1.81, $p = 0.37$; non-LOF HR = 1.32, 95% CI 0.92,1.9, $p = 0.13$) (Supplementary Table 1). Results were similar when combining PBRM1 and ARID2; LOF HR = 0.85, 95% CI 0.65, 1.1, $p = 0.25$ unadjusted and HR = 1.1, 95% CI 0.83,1.45, $p = 0.52$ adjusted. Given the higher frequency of ARID2 mutations in the non-RCC cohorts, we combined PBRM1 and ARID2 LOF and non-LOF mutations for individual subtype analysis, which was significant in non-small-cell lung cancers (Fig. 6a, Supplementary Table 2). When univariately examining LOF mutations in PBRM1 and ARID2 as well as LOF in PBRM1 alone, they remained significantly associated with adverse OS in non-small-cell lung cancer (Fig. 6b, c). In individual cancer types, PBRM1 was correlated with worse OS in non-small-cell lung cancers ($n = 983$; HR 2.91, $p < 0.001$) after adjusting for TMB and total CNA (Supplementary Table 3). A

significant correlation with adverse OS was also seen in bladder cancer ($n = 245$; HR 11.85, $p < 0.001$); however, only three PBRM1 mutants comprised this group. ARID2 was not significant in either cancer type.

**PBRM1 mutations and the TME.** Previous work by our group and others suggested that PBRM1 loss was associated with further hypoxic signaling and angiogenic expression[11, 12]. This was further bolstered by the association with improved response of PBRM1 mutated tumors to VEGF blockade therapies[13–15]. We utilized transcriptomic data from three independent cohorts to analyze the impact of PBRM1 LOF mutations on transcriptional pathway enrichment. These included COMPARZ[16], a phase 3 randomized trial comparing the efficacy and safety of pazopanib and sunitinib as first-line therapy ($n = 352$ (targeted exome and whole-genome RNA microarray)), McDermott et al.[4], a randomized phase 2 study of atezolizumab alone or combined with bevacizumab versus sunitinib in treatment-naive metastatic renal cell carcinoma ($n = 201$ (whole exome + RNASeq)), and Miao et al.[8], which analyzed a cohort of approximately 100 metastatic ccRCC patients to identify genomic alterations correlating to response to ICB ($n = 41$ (whole exome + RNASeq)). All three cohorts demonstrated higher hypoxia pathway enrichment in PBRM1 mutated samples with GSEA $p$ value as 0.002, 0.008, and 0.002, respectively. In the COMPARZ and McDermott et al. cohorts, we observed downregulation of interferon alpha and gamma response genes. With respect to interferon gamma response or JAK/STAT signaling, we were able to validate higher expression in the Miao et al. cohort (as previously reported) but we found lower expression in both the COMPARZ and McDermott et al. cohorts. We further performed immune deconvolution using ssGSEA focusing on immune and angiogenic gene signatures. We consistently observed significantly higher angiogenic gene expression in PBRM1 mutated tumors in the COMPARZ and McDermott et al. data sets, $p = 0.0004$ and 0.005, respectively, and a similar trend in Miao et al. cohort (Fig. 7a). Further, immunohistochemical (IHC) staining from the COMPARZ and McDermott cohort demonstrates significantly higher CD31-positive staining in PBRM1 mutated tumors, implying higher

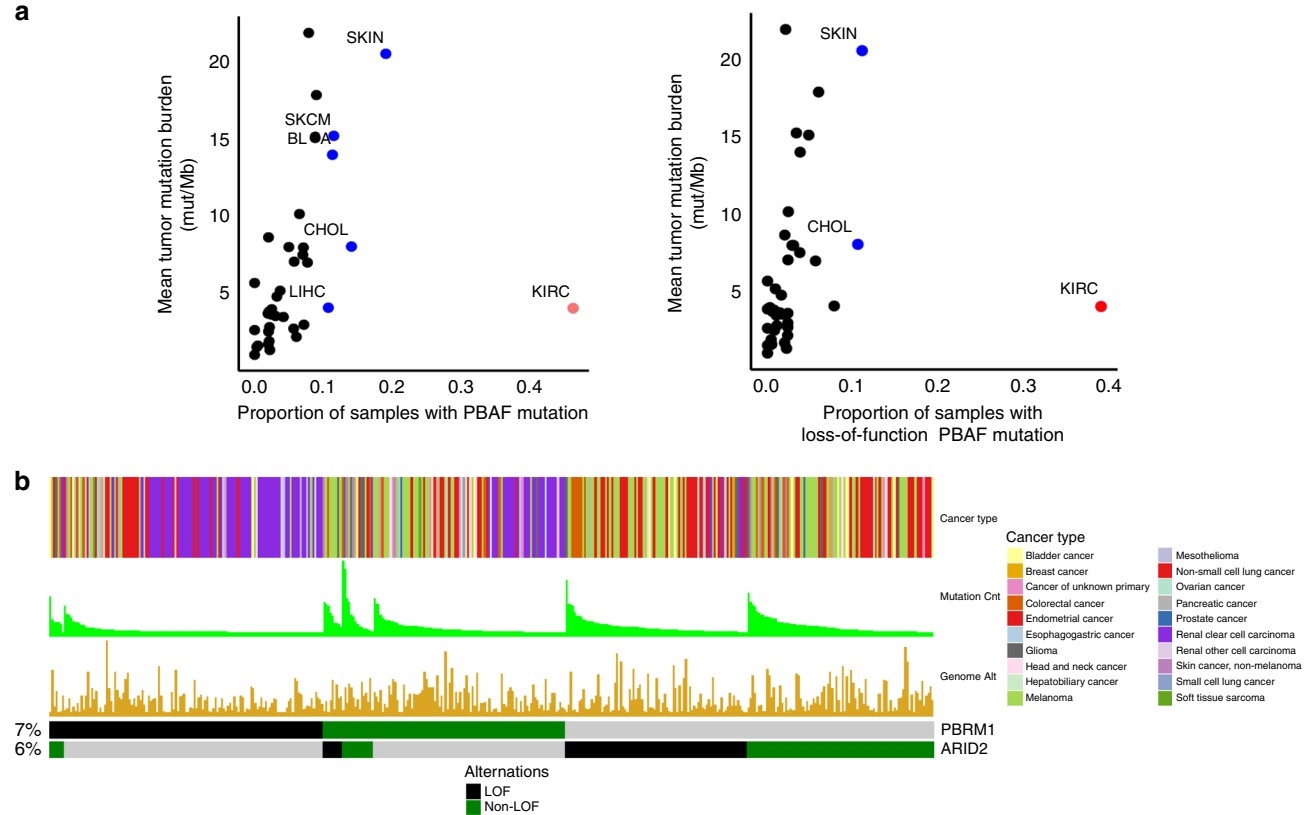

**Fig. 3 PBAF (*PBRM1* and *ARID2* only) complex mutations across MSK-IMPACT (*n* = 3700). a** All PBAF complex mutations as a function of mean tumor burden (left) and loss of function (LOF) only mutations (right). **b** OncoPrint plot demonstrating loss-of-function vs. non-loss-of-function PBAF complex mutations across MSK-IMPACT.

| **Table 2 Characteristics of 189 patients with clear cell RCC treated with ICB therapies.** | |
|---|---|
| | **All (*n* = 189)** |
| Age at treatment (years)—median (range) | 60 (34, 89) |
| Sex | |
| Male | 142 (75%) |
| Histology subtype | |
| Clear cell RCC | 189 (100%) |
| IMDC risk score at starting ICB therapy | |
| Good | 54 (29%) |
| Intermediate | 102 (54%) |
| Poor | 24 (13%) |
| Missing | 9 (5%) |
| ICB therapy type | |
| Single-agent IO | 75 (40%) |
| IO + IO combination | 38 (20%) |
| IO + VEGF combination | 71 (38%) |
| IO + other treatment combination | 5 (3%) |
| Line of ICB therapy | |
| First line | 97 (51%) |
| ≥Second line | 92 (49%) |
| *PBRM1* mutation type | |
| LOF | 61 (32%) |
| Non-LOF | 27 (14%) |

*ICB* immune-checkpoint blockade, *IMDC* International Metastatic Renal Cell Carcinoma Database Consortium, *LOF* loss of function.

degrees of tumor angiogenesis in *PBRM1* mutated tumors (Fig. 7b). IHC studies from the two cohorts also reveal higher PD-L1-negative and lower PD-L1-positive staining tumors in *PBRM1* mutated tumors compared to wild type (Fig. 7b) and no difference in CD8 positivity between *PBRM1* mutant and wild-type tumors (Supplementary Fig. 3). Immune deconvolution of bulk expression data failed to find any specific immune enrichment patterns across the three cohorts when stratified by *PBRM1* mutation status (Supplementary Fig. 4).

## Discussion

The identification of genomic biomarkers for ICB therapy remains an evolving field. While several studies seem to validate tumor mutation and neoangtigen burden, along with mismatch repair mutations[1–3], numerous other studies have relied on single gene mutations with relatively small cohorts and often not correcting for potential confounding factors such as TMB or microsatellite instability/mismatch repair status. Our analysis of over 3000 patients treated with ICB did not find an association with PBAF complex loss and ICB response in both univariate or multivariate tests. Specifically, we found PBAF complex mutations to be most common in ccRCC tumors which were dominated by *PBRM1* mutations; interrogation of this cohort of ICB-treated ccRCC patients failed to reveal an association between *PBRM1/ARID2* mutations and overall survival or time-to-treat failure. Furthermore, analysis of both the McDermott et al. and our recently analyzed COMPARZ cohort showed unchanged or lower IFNγ signaling in the *PBRM1* mutants.

3p loss (which encompasses four commonly mutated genes: *VHL, PBRM1, SETD2, BAP1*) is a ubiquitous, pathognomonic event in ccRCC, occurring in upwards of 90% of tumors. Preclinical data support the notion that *VHL* (the most commonly mutated gene in clear cell RCC) and *PBRM1* co-occur; although *VHL* is the initial driver event in the pathogenesis of clear cell RCC, genetic deletion of *VHL* in mice is insufficient to initiate kidney tumors[12]. After loss of *VHL*, loss of additional 3p21 tumor suppressor genes (*PBRM1*) further activates HIF1/STAT3 signaling in mouse kidney and positions mTORC1 activation as the third preferred driver event.

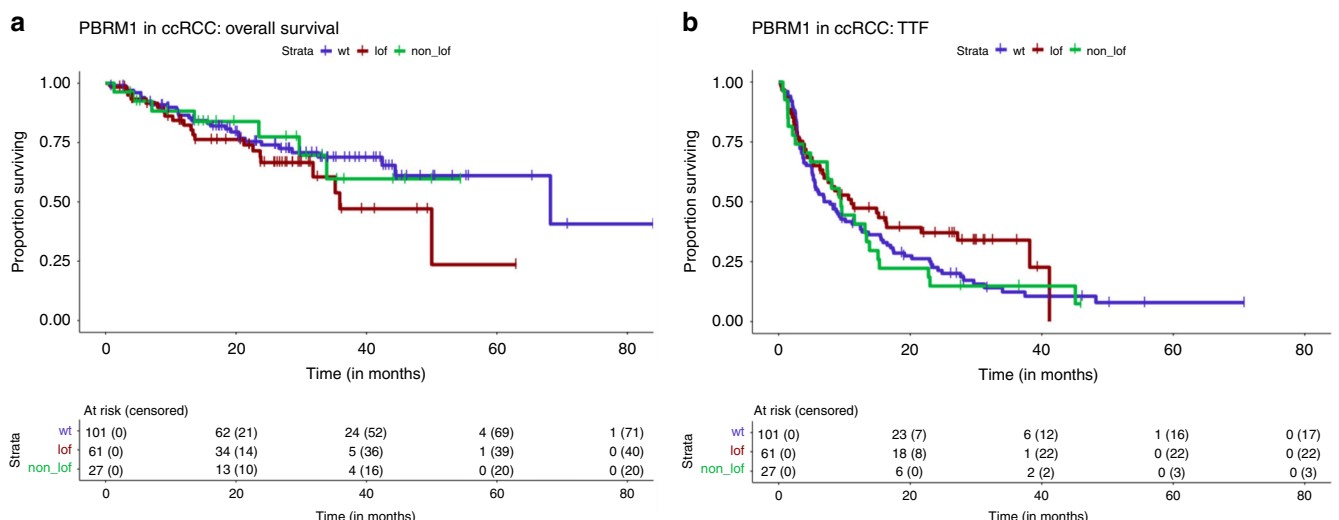

**Fig. 4 Survival and time to treatment failure in PBRM1 mutated MSKCC ccRCC patients.** Kaplan–Meier curves demonstrating overall survival (median 68.2 months; 95% CI 44.4, NA) and time-to-treatment failure (TTF) (median 8.9 months; 95% CI 6.9, 12.42) in clear cell RCC patients across MSK-IMPACT.

*PBRM1* encodes for the protein BAF180, a subunit of the PBAF subtype SWI/SNF chromatin remodeling complex. The PBAF complex regulates access to DNA bound to histones by transiently altering the nucleosome structure so that the DNA may be accessed by the cell's transcriptional machinery. Mutation of *PBRM1* leads to the integration of an altered BAF180 subunit and thereby alters the normal activity of the PBAF complex.

The PBAF complex has been reported to control the immune system by modulating immune recruitment and T cell activation through the IFNγ signaling and JAK/STAT pathway. IFNγ serves as a critical cytokine for tumor immunity[17] and its signaling axis is mediated through STAT3 and STAT5 signal transduction[18]. Notably, STAT3 is part of an important intrinsic pathway for inflammation by inducing genes that lead to the production of IL-6, 10, 11, 17, 23, CXCL12, and COX-2 (ref. [19]). The activation of STAT3 is normally a transient process activated by numerous cytokines, including IL-6 (ref. [20]), IFNγ[21], and TNFa[22], which is strictly controlled to prevent unscheduled gene regulation. The pathway of activation is initiated through JAK phosphorylation whereby STAT3 becomes phosphorylated[23] and combines into dimers to enter the nucleus via importin alpah5/NPI-1 (ref. [24]). Once in the nucleus, STAT3 induces transcription of genes involved in cell survival and proliferation[25, 26].

However, unbridled activation of STAT3 is oncogenic[27] and has been reported in a variety of tumors[28], including renal cancers[29], and can go unchecked due to mutations in negative regulatory mechanisms[30]. Growing evidence supports the role of *PBRM1* (via PBAF) as one of these negative regulators of STAT3 and in turn acts to downregulate the transcription of proliferative and interferon stimulated genes (ISG). In 2011, Verela et al.[31] showed that *PBRM1* knockdown enhanced proliferation and migration of kidney cancer cell lines. Later, Pan et al.[7] showed through RNA-sequencing that among *PBRM1*-deficient cells, gene sets related to IFNγ and IFNα response were significantly enriched compared to controls when treated with IFNγ. Moreover, the authors showed that cells deficient in *PBRM1* secreted larger amounts of chemokines necessary for the recruitment of effector T cells compared to controls following IFNγ stimulation[7] and that mRNA levels of *PBRM1* negatively correlated with expression of granzyme B and Perforin 1, as well as with the granzyme B/CD8A ratio; all this suggests that a lower expression of *PBRM1* correlated to higher cytotoxic T cell activity and that

*PBRM1* status may be relevant for immune activation and infiltration, perhaps due to STAT3 ISGs[7]. Similarly, Miao et al.[6] showed genes which were most strongly enriched in PBAF knockout cell lines were immune stimulatory. Indeed, transcription analysis of DEG between *PBRM1* mutant and wild types in the Miao et al. cohort supported increased IFNγ gene expression. However, our analysis of two significantly larger metastatic ccRCC cohorts, COMPARZ and McDermott et al., demonstrated lower IFNγ and JAK/STAT3 expression. Given the tight counterbalance between the PBAF complex and ISGs it is highly conceivable that additional alteration can shift the balance away from ISG signaling and can explain the lack of efficacy to ICB seen in an array of PBAF complex mutated cancers.

Furthermore, there has been evidence to suggest that the HIF pathway, the main driver of ccRCC in VHL loss, itself may be associated with the immune response through *PBRM1*. The PBAF complex has been shown to suppress the hypoxia transcriptional signature in VHL negative ccRCC[11, 12]. Nargund et al.[12] showed through GSEA analysis in untreated ccRCC from the TCGA and a *PBRM1* knockout mouse model that *PBRM1* loss had increased transcriptional outputs of HIF1 and STAT3. The authors show HIF1α and STAT3 cooperate to activate the expression of HIF1α targets including genes involved in angiogenesis through a feed-forward amplification loop[32]. It has been separately shown that HIF1α induces PKM2 to activate STAT3 which prompts further HIF1α expression[33, 34]. Similar findings were reported by Miao et al.[6] after GSEA analysis of RNA-sequenced pre-treated ccRCC *PBRM1* LOF tumors which showed increased expression of hypoxia and IL-6/JAK-STAT3 gene sets. If *PBRM1* acts to inhibit STAT3, the inference is that *PBRM1* prevents the amplification of the HIF1α and STAT3 transcriptional outputs upon VHL loss[12]. However, when *PBRM1* is also lost through mutation, the intrinsic HIF1α-STAT3 propagation goes unchecked. Indeed, Nargund et al.[12] showed that in VHL knockout cells, *PBRM1* suppresses the self-perpetuating amplification of HIF1α/STAT3 signaling. Consistently, we found higher angiogenic expression in *PBRM1* mutated tumors. Clinically, this is supported by the fact that *PBRM1* mutated tumors respond better to antiangiogenic therapy. Carlo et al.[35] demonstrated in 105 patients with metastatic ccRCC who had received systemic therapy, TTF with VEGF-targeted therapy differed significantly by *PBRM1* mutation status, where *PBRM1* mutants associated

**Table 3 Univariate and multivariate regression models of ICB and combination therapy response in *PBRM1* mutated ccRCC patients in MSKCC cohort (*n* = 189).**

| Variable | Time-to-treatment failure | | | | | Overall survival | | | | |
|---|---|---|---|---|---|---|---|---|---|---|
| | Univariate analysis | | Multivariate model | | | Univariate analysis | | Multivariate model | | |
| | HR (95% CI) | P value | HR | 95% CI | P value | HR (95% CI) | P value | HR | 95% CI | P value |
| *PBRM1* wild type | 1 | | Ref. | | | 1 | | Ref. | | |
| LOF | 0.73 (0.5–1.07) | 0.112 | 0.85 | 0.57, 1.28 | 0.44 | 1.5 (0.85–2.66) | 0.161 | 1.24 | 0.69, 2.25 | 0.47 |
| Non-LOF | 1.05 (0.67–1.65) | 0.838 | 1.22 | 0.77, 1.94 | 0.4 | 1.05 (0.46–2.4) | 0.914 | 0.88 | 0.36, 2.14 | 0.78 |
| TMB | 0.94 (0.89–0.99) | 0.029 | 0.96 | 0.90, 1.02 | 0.19 | 0.99 (0.9–1.08) | 0.778 | | | |
| Age at treatment | 1.29 (0–610.11) | 0.936 | | | | 1.31 (0–5668.63) | 0.96 | | | |
| Genome doubled | 1.08 (0.74–1.59) | 0.695 | | | | 1.56 (0.89–2.74) | 0.115 | | | |
| Fraction CNA | 1.46 (0.79–2.71) | 0.227 | | | | 2.04 (0.78–5.35) | 0.144 | | | |
| CTLA-4 status | | 0.155 | | | | | 0.376 | | | |
| No | 1 | | | | | 1 | | | | |
| Yes | 1.33 (0.9–1.99) | | | | | 0.74 (0.38–1.44) | | | | |
| IMDC risk | | 0.211 | | | | | <0.001 | | | |
| 1, 2 | 1 | | | | | 1 | | | | |
| 3 | 1.35 (0.84–2.17) | 0.003 | | | | 3 (1.59–5.66) | | 4.22 | 2.18, 8.17 | <0.001 |
| Drug class | | | | | <0.001 | | 0.028 | | | |
| IO | 1 | | 1 | | | 1 | | | | |
| IO–IO | 1 (0.66–1.54) | | 1 | | | 0.54 (0.27–1.09) | | | | |
| IO–VEGF | 0.55 (0.38–0.8) | | 0.54 | 0.38, 0.76 | | 0.46 (0.24–0.87) | | | | |
| Line of therapy | | 0.211 | | | | | <0.001 | | | |
| >1 | 1 | | | | | 1 | | | | |
| 1 | 0.81 (0.59–1.12) | | | | | 0.34 (0.19–0.59) | | 0.27 | 0.15, 0.49 | <0.001 |
| BAP1 | | 0.842 | | | | | 0.702 | | | |
| No | 1 | | | | | 1 | | | | |
| Yes | 0.96 (0.64–1.44) | | | | | 1.14 (0.59–2.21) | | | | |
| SETD2 | | 0.48 | | | | | 0.71 | | | |
| No | 1 | | | | | 1 | | | | |
| Yes | 0.88 (0.62–1.26) | | | | | 0.9 (0.51–1.59) | | | | |

P values derived from Cox proportional hazards model. For multivariate model IO and IO/IO were combined into one category. Multivariate model is based on 180 patients with available risk score.

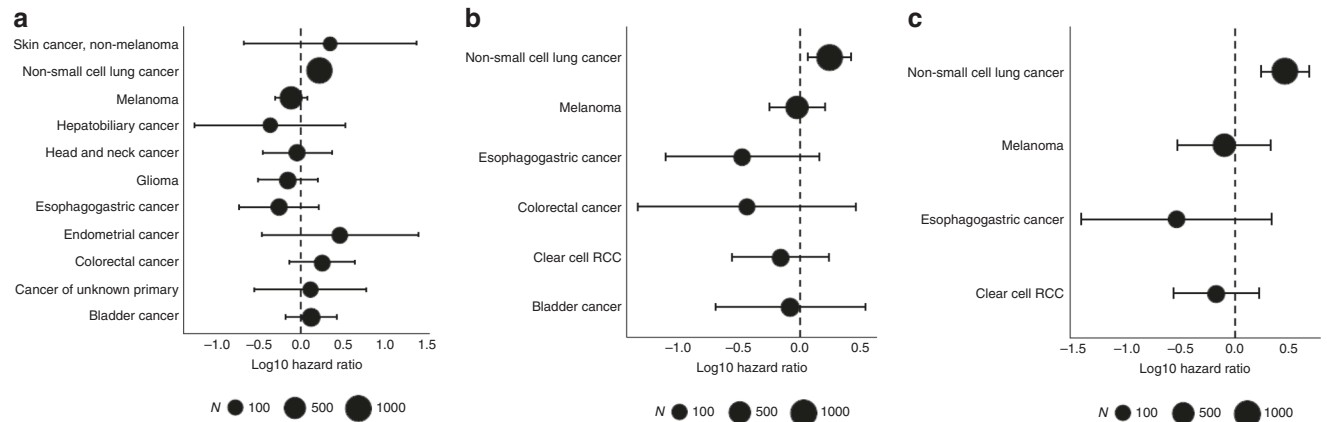

**Fig. 5 Survival and time to treatment failure in PBRM1 mutated MSKCC ccRCC patients by line of therapy.** Overall survival (OS) and time-to-treatment failure (TTF) in MSK-IMPACT ccRCC ($n = 173$) stratified by **a** first line and **b** ≥second line of treatment (line of treatment not available in 12/185 patients).

**Fig. 6 Forest plots of overall survival by PBRM1 or ARID2 loss.** Forest plots demonstrating hazard of death in ICB-treated patients examining **a** *PBRM1* or *ARID2* LOF + non-LOF mutations, **b** *PBRM1* or *ARID2* LOF mutations alone, and **c** *PBRM1* LOF mutations alone. Error bars represent 95% confidence interval.

with more favorable TTF ($p = 0.01$, median 12.0 months for *PBRM1* mutants versus 6.9 months for wild-type tumors)[35].

Finally, we have shown in the COMPARZ cohort that *PBRM1* mutations are associated with both higher angiogenesis expression and response to anti-VEGF therapy. This parallels the findings of the sunitinib arm of the McDermott et al. cohort,

wherein sunitinib efficacy was enriched in highly angiogenic tumors and coincided with *PBRM1* mutant tumors, demonstrating improved progression free survival and objective response rates in tumors with high angiogenic gene signature.

Braun et al.[9] recently reported on the association between *PBRM1* alterations and ICB response in CheckMate 025, a large

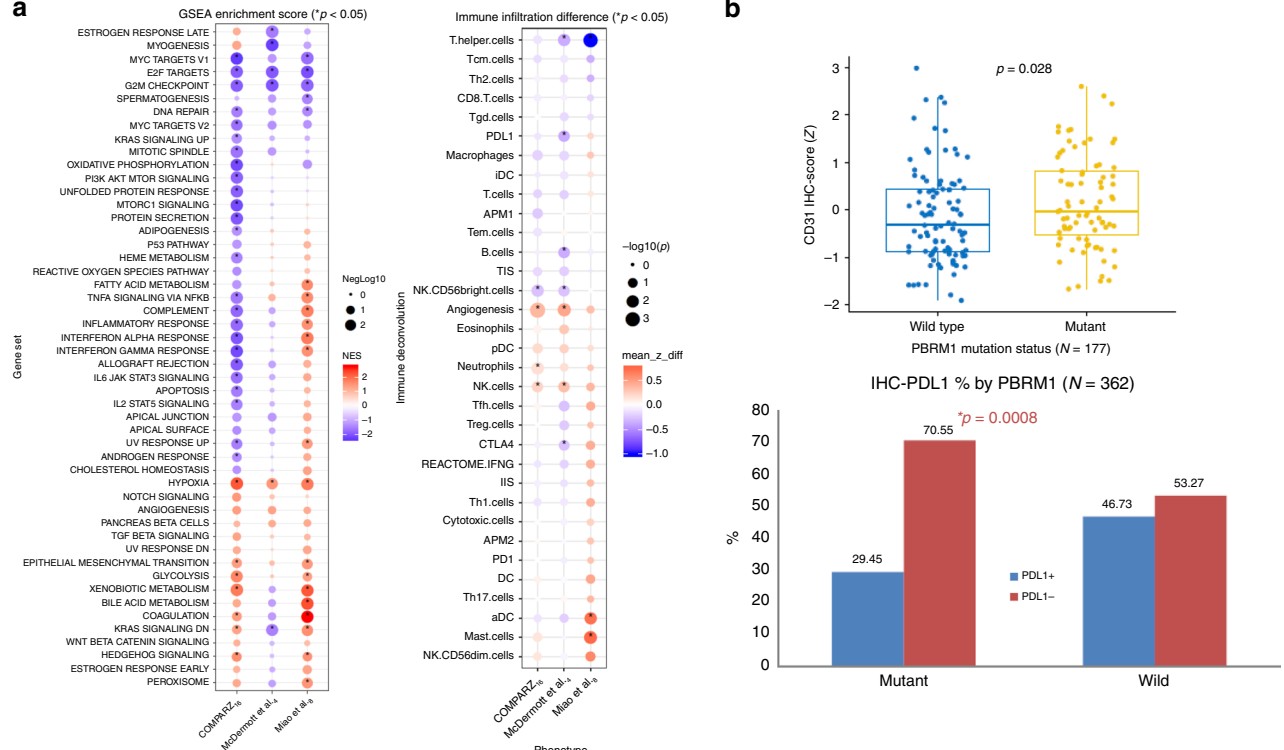

**Fig. 7 Immune deconvolution of PBRM1 mutated tumors. a** Immune deconvolution using single sample gene set enrichment analysis (GSEA), focusing on immune and angiogenic gene signatures. Significantly higher angiogenic gene expression was observed in *PBRM1* mutated tumors in the COMPARZ[16] and McDermott et al.[4] data sets, p = 0.0004 and 0.005, respectively, and a similar trend in the Miao et al.[8] cohort. **b** Immunohistochemistry staining results from COMPARZ[16] and McDermott et al.[4] data sets demonstrate significantly higher CD31+ staining in *PBRM1* mutated tumors and lower PD-L1+ staining in *PBRM1* mutated tumors. Box plot: middle line of box indicates median and the bounds indicate quartile 1 and quartile 3. The whiskers reach to the maximum/minimum point within the 1.5 × interquartile range from quartile 3/quartile 1, respectively. P values from COMPARZ[16] bar plots generated by Fisher's exact test; p values from the GSEA plot derived from a permutation test; p values from immune deconvolution difference plot and box plots derived from Wilcoxon rank-sum test. The Fisher's Exact test and Wilcoxon rank-sum test p values are two sided. No adjustments made for multiple comparisons; all p values are nominal.

randomized phase 3 trial of nivolumab versus everolimus in advanced renal cell carcinoma[10]. The validation study demonstrated a modest, albeit significant, mutation effect on improved response and survival in nivolumab-treated patients, none in subjects on the everolimus arm. Intriguingly, this effect was observed in patients who received prior antiangiogenic therapy; previous studies of *PBRM1* mutations in the first-line setting had negative results, and *PBRM1* alterations have also been associated with benefit from antiangiogenic therapies[4, 6]. It is noteworthy to mention that among nivolumab-treated patients, a higher proportion of responders (15 of 38) harbored truncating *PBRM1* mutations, which was statistically significant but numerically similar to the non-responders (16 of 74). This lack of effect in the first-line setting may explain the lack of response of the atezolizumab arm in the *PBRM1* LOF mutations in the McDermott et al. cohort, a finding we were also unable to validate in both first- or second-line patients.

As the field of precision oncology continues to evolve, limitations to its current applicability in the clinical setting exist. Novel putative prognostic or predictive molecular biomarkers should prove complementary or superior to the best available clinical prognostic or predictive factors before entering clinical practice. Indeed, prior studies have specifically examined the applicability of genomic signatures to improve the prognostic performance of established prognostic models; in one such instance, de Velasco et al.[36] demonstrated a model of genomic signatures which improved the prognostic performance of IMDC and MSKCC risk classifications. Globally, factors predicated around the cost of large-scale genomic analyses, fostering collaborative efforts on the behalf of dedicated independent research programs so as to limit waste of resources, and tumor-specific factors such as intratumor heterogeneity, must be brought to the forefront of any discussion regarding quests for novel molecular biomarkers[37].

## Methods

**Cohort selection.** *TCGA:* Pan-cancer TCGA cohort (n = 10,359 from 31 cancer subtypes) data, including *ARID2, BRD7,* and *PBRM1* mutations, mutation count as well as FGA (which is an estimate proportion of the tumor genome affected by copy number gains and losses), were queried and downloaded from MSKCC cBioPortal (cbioportal.mskcc.org). Specifically, FGA is calculated as the length of segments with log2 CNA value larger than 0.2, divided by the length of all segments measured. Among various mutation types the truncating mutation (putative driver) is treated as a LOF mutation and the remainder treated as non-LOF.

*MSK ICB Cohort:* After receiving institutional review board (IRB) approval at MSKCC, institutional pharmacy records were used to identify patients who had received at least one dose of immunotherapy at MSKCC for metastastic cancer, and these were then cross-referenced with patients who had MSK-IMPACT testing done in the context of routine clinical care. Informed consent was obtained from all patients prior to MSK-IMPACT testing. For MSK pan-cancer cohort we selected all patients treated with immunotherapy at MSKCC between 2010 and 2018 who had their tumor profiled with MSK-IMPACT targeted sequencing platform (n = 3700 from 51 cancer subtypes). After limiting non-renal cell carcinoma patients to 11 cancer types with at least 50 patients and 5 *PBRM1* or *ARID2* mutations, the pan-cancer cohort was comprised of 2936 patients.

Details of tissue processing and next-generation sequencing and analysis were previously described[38]. Patients enrolled in ongoing clinical trials for which study outcomes have not been reported were removed, as were a small proportion of patients with either localized disease treated in the neoadjuvant setting or localized

disease. Other preceding or concurrent non-ICB treatments were not recorded or accounted for in the analysis.

**IMPACT mutational profiling**. The MSK-IMPACT assay was performed on DNA extracted from formalin-fixed, paraffin-embedded primary tumor samples as previously published[39]. The total number of somatic mutations identified was normalized to the exonic coverage of the respective MSK-IMPACT panel in megabases. Importantly, concurrent sequencing of germline DNA from peripheral blood was performed for all samples to identify somatic tumor mutations. For each histology, we subsequently identified cases in the top twentieth percentile of TMB and determined the log-rank $p$ value for difference in overall survival (OS) and the direction of the effect with an HR determined from a coxph model. Additional analyses were performed with the TMB cutoff ranging from 10 to 50%, as well as with the TMB cutoff instead defined among all patients (both ICB- and non-ICB-treated patients). LOF mutations were defined as any truncating mutation (non-sense, frameshift) or homozygous deletion. Splice site mutations were not considered LOF mutations as it is not determinable from our data if in fact mutations at splice sites cause frameshifts or premature stops resulting in LOF, or perhaps activate cryptic splice sites instead.

**Statistical methods**. OS was defined as time from the start of immunotherapy until death or last date of follow-up. For patients who received multiple courses of ICB, the first treatment was used for analysis. Patients were censored at the date of most recently attended appointment at MSKCC if death was not recorded in the electronic medical record. For the ccRCC cohort, TTF was defined as time from the start of immunotherapy until treatment discontinuation for any reason. The Kaplan–Meier method was used to estimated TTF and OS curves. Effect of the mutations was tested using univariate and multivariate Cox proportional hazard regression. For the analysis of pan-cancer cohort, the Cox model was stratified by the cancer type. In analysis of ccRCC cohort all variables with $p < 0.05$ univariately were included in the multivariate model. Wilcoxon rank-sum test implemented as R wilcox.test function was used for testing the difference of deconvolved immune features between mutant and wild-type groups. All analyses were performed in the R platform v3.6.1.

**Reporting summary**. Further information on research design is available in the Nature Research Reporting Summary linked to this article.

## Data availability

Sequencing data from the MSK-IMPACT and TCGA cohorts were obtained from MSKCC cBioPortal (http://cbioportal.org) and the GDC data portal (https://gdc.cancer.gov/about-data/publications/pancanatlas), respectively. The expression data used for the transcriptomic analyses were obtained from the original publications and are publicly available (McDermott et al.[4] cohort: European Genome-Phenome Archive (EGA) at accession number EGAS00001002928, Miao et al.[8] cohort: dbGap at accession number phs001565.v1.p1). Trial data sets for COMPARZ[16] are available on application to the linked Data Access Committee [novartis.datasharing@novartis.com] at the EGA repository under the accession code EGAD00010001930. All remaining relevant data are available in the article, Supplementary Information, or from the corresponding author upon reasonable request.

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

## Acknowledgements

We express our deep gratitude to our patients, their families, and their caregivers. We acknowledge support from the Cancer Center Support Grant of the National Institutes of Health/National Cancer Institute (P30CA008748), the Weiss family funds, and the Ruth L. Kirschstein National Research Service Award (T32CA082088).

## Author contributions

A.A.H., M.H.V., T.A.C., and R.J.M. conceived and supervised the project. A.A.H. and K.A. wrote the manuscript with input from all authors. I.O., J.F., F.K., R.G.D., and E.R. were involved in formal data analysis and supervised the methodology of the project. D.H., K.A.B., Y.G., and S.M.K. curated the data. A.B., J.H., and C.J.F. provided data needed for analysis. All authors proofread the manuscript.

## Competing interests

A.A.H. corresponding author, certifies that all conflicts of interest, including specific financial interests, relationships, and affiliations relevant to the subject matter or materials discussed in the manuscript are the following: T.A.C. is a paid consultant for Bristol-Myers Squibb and Illumina, reports receiving commercial research grants from Astra-Zeneca, Illumina, Pfizer, and BMS, and holds ownership interest (including patents) in Gritstone. M.H.V. reports honoraria from Novartis; consulting/advisory role for Alexion Pharmaceuticals, Bayer, Calithera Biosciences, Corvus Pharmaceuticals, Exelixis, Eisai, GlaxoSmithKline, Natera, Novartis, and Pfizer; research funding from Pfizer, Bristol-Myers Squibb, and Genentech/Roche; and travel, accommodations, and expenses from Eisai, Novartis, and Takeda. R.J.M. reports grants and personal fees from Pfizer, Novartis, Eisai, Genentech/Roche; personal fees from Exelixis, Merck, Lilly, and Incyte; and grants from BMS outside the submitted work. The remaining authors declare no conflict of interest.
