## [Peer Review File · Nature Communications]

REVIEWERS' COMMENTS (FIRST ROUND OF REVIEW):

Reviewer #1 (Remarks to the Author):

The quest for reliable and reproducible biomarkers of immunotherapy activity in several tumor types is one of the "hottest" fields of research in Oncology, albeit presently available results are too often inconclusive or - even worse - conflicting. Recently, an association between mutations in the SWI/SNF chromatin remodeling complex and response to immune checkpoint inhibitors has been proposed by some Authors (especially in clear cell kidney cancer), although once again other Researchers have not fully confirmed these observations. In this elegant paper, the Authors checked for the frequency of PBAF mutations in two large pan-cancer series, explored the prognostic significance of PBAF mutations across a number of different malignancies, and assessed the impact of PBRM1 Loss Of Function mutations on tumor microenvironment expression programs in a cohort of clear cell kidney cancer patients.

Overall, the paper is well written and, in my opinion, appears extremely sound from a methodological viewpoint. However, the discussion on the clinical implications of these data, within the context of other already available data, could be improved.

MINOR POINTS

1. Authors should discuss the concept that any novel putative prognostic and/or predictive molecular biomarker (the latter being definitely more important) should prove to be superior to the best available clinical prognostic/predictive factor, before entering clinical practice. This is even more relevant if the proposed molecular biomarker is much more complex, costly and difficult to be reproduced, as compared to the clinical one. If the above is true, it is clear that we are still in the infancy of our quest for a really useful molecular biomarker and this should be clearly highlighted.

- *We appreciate the reviewers' comments and certainly agree with the notion that novel prognostic and/or predictive molecular biomarkers should prove to be superior to the best available clinical prognostic/predictive factors before entering clinical practice. In particular, two recent studies have specifically examined the applicability of genomic signatures to improve the prognostic performance of established prognostic models; de Velasco et al demonstrated a model of genomic signatures which improved the prognostic performance of the IMDC and MSKCC classification [de Velasco et al, Oncologist, 2017]. Similarly, a study from our institution reported an integrated genomic and transcriptomic analysis of patients with clear cell renal cell carcinoma and identified four distinct molecular subgroups associated with response and survival, which ultimately has broad implications for optimizing precision treatment of renal cell carcinoma [Hakimi et al, Cancer Discov, 2019]. These relevant points have been added to the discussion section of the manuscript (changes highlighted in yellow).*

2. Theoretically, the most important use of a molecular/genetic biomarker would be to aid us in selecting the treatment most likely efficacious for a given patient with a given tumor, or else to avoid potentially unuseful and/or harmful treatments. A number of preliminary reports have been used in the past few years to claim for a precision approach to cancer treatment. Given the

conflicting results available, and the lack of useful correlations emerging from important studies such the present, despite intriguing and promising backgrounds, Authors should briefly discuss the present limits of precision Medicine. In this sense, commenting the words of wisdom from Tannock and Hickman (N Engl J Med 2016;375:1289-94) would be appropriate.

- ***We agree with the reviewers' point in the importance of recognizing the current limitations of precision medicine, and as such, these limitations are included and expounded upon in the discussion section of the manuscript.***

Reviewer #2 (Remarks to the Author):

Hakimi et al. assessed the frequency of PBAF complex mutations across two large pan-cancer cohorts (TCGA and MSKCC), they evaluated the prognostic significance of PBRM1 mutations in clear cell RCC patients, and examined the association of PBRM1 mutations with clinical response to ICB therapies across a series of solid tumor cohorts from MSKCC. The authors leveraged their access to the large genomic and clinical trial datasets to address an important question. While this work is of interest, there are however a number of issues (e.g superficial analysis, lack of mechanistic insights) that limit the overall impact of the study.

Specific points:

1. Only three (two for the ICB studies) genes of the PBAF complex were studied, while other potentially functional elements of this complex were not included.

- ***Of the three genes, we studied PBRM1 and ARID2 in our MSKCC pan-cancer and RCC cohort; BRD7 is not included in MSK-IMPACT, and was therefore unable to be studied (this is mentioned in paragraph two of the results section).***

2. Loss of function of PBRM1 was solely defined based on the single platform mutation data (whether there was frameshift or nonsense mutations in this gene. It is unclear whether the authors included the splicing site mutations as LOF or classified them as non-LOF mutations; Were the missense mutations treated uniformly as non-LOF without further functional evaluation? The clonal status of the mutations were not analyzed. It is unclear whether the author stratified their analysis by copy number alterations (such as 3p loss which is very frequent in some cancers); Importantly, other alterations such as RNA editing, splicing variants, or epigenetic dysregulation may also lead to loss of function of PBAF complex, but was not examined in this study.

- ***The reviewer brings to light many salient points. To address the first point, splice site mutations were not considered LOF mutations; although splice site mutations sometime do cause LOF (frameshift or premature stop), sometimes they do not. They may instead activate cryptic splice sites, and the difference between such cases is not determinable from our data. This has now been added to the methods section. Additionally, to address this point, we've repeated our analysis to include splice site mutations as LOF to determine how, if any, changes this makes to our results. RCC***

COHORT: among 189 ccRCC patients, 11 patients have splice mutations and get reclassified as LOF, and none of the ARID2 mutations are reclassified. In multivariate models of ccRCC patients treated with ICB (n=189), LOF mutations in PBRM1 were not associated with OS (HR=1.19, p=0.56), change from HR=1.24, p=0.47 when NOT considering splice mutations as LOF. Similarly, there was no effect on time to treatment failure in ccRCC (HR=0.89, p=0.56), change from HR=0.85, p=0.44. PAN-CANCER COHORT: In a series of 11 other solid tumors with a minimum of 50 patients and 5 PBRM1/ARID2 mutations each (n=2,936), only 7 splice site mutations get reclassified as LOF. LOF mutations were not associated with improved OS in a stratified multivariate model, HR=1.37, p=0.1, change from (LOF HR=1.2 p=0.35). In summary, our analysis considering splice site mutations as LOF does not alter our conclusions or final results in any meaningful direction.

Missense mutations were treated uniformly as non-LOF, but certainly agree that some missense mutations may lead to LOF. To address the point that the clonal status of mutations were not analyzed, several high-impact sequencing studies including TRACERx (Turajlic et al, Cell, 2018) and TCGA (Creighton et al, Nature, 2013) have shown that 3p loss (which encompasses four commonly mutated genes: VHL, PBRM1, SETD2, BAP1) is a ubiquitous, pathognomonic event in clear cell RCC, occurring in upwards of 90% of tumors. This point has been included in the methods section (highlighted in yellow).

3. It is unclear about the functional impact of the genomic mutations, as LOF of PBAF complex was not assessed at protein level.

- **The reviewer brings to light an important point regarding the functional impact of LOF mutations of the PBAF complex. While we did not test protein expression levels, immunohistochemical data from others suggest that protein level loss is similar to PBRM1 mutation rate in clear cell RCC, approximately 43% (Ho et al, Urol Oncol, 2015).**

4. The tumors that were wt or non-lof for PBRM1 may carry other genomic or epigenomic alterations that result in PBRM1 loss.

- **We agree with the possibility that wild-type or non-lof for PBRM1 may carry other genomic or epigenomic alterations that result in PBRM1 loss.**

5. For the survival analysis, the authors should discuss the potential confounding factors (clinical, histopathological, or genomic that are likely to also confound the analysis), and if needed, stratify their analysis by the confounding factors.

- **We agree that there may be potential confounding factors in our analysis, and as such, we have stratified our analysis by the potential confounding factors found in the regression models in Table 3.**

6. For the correlation analysis with ICB response, the authors should include other known factors

that influence patient response such as the levels of immune cell infiltration, CD8 expression, cytotoxic T cell proportion, PD-1/PD-L1/CTLA-4 expression, etc. and correlate these known factors with PBAF complex status to better understand their function in immune regulation and predictive/prognostic values on ICB response.

- ***Data from others, including secondary analyses of the Checkmate 025 clinical trial [Motzer et al, N Engl J Med, 2015], has demonstrated that PD1/PDL1 status in RCC has not been shown to correlate with response to ICB. We unfortunately do not have RNA data for our cohort of patients, as RNA data is not included in MSK-IMPACT. We correlated many of the factors that the reviewer mentions with PBRM1 status, which is found in Figure 6. We have now added to our results the immunohistochemical (IHC) staining from the COMPARZ and McDermott cohorts which demonstrate significantly higher CD31 positive staining in PBRM1 mutated tumors, implying higher degrees of tumor angiogenesis in PBRM1 mutated tumors (Figure 6B). IHC studies from the two cohorts also reveal higher PDL1 negative and lower PDL1 positive staining tumors in PBRM1 mutated tumors compared to wild-type (Figure 6B) and no difference in CD8 positivity between PBRM1 mutant and wild-type tumors (Supplementary Figure 3).***

7. The patients may have a heterogeneous background (if ICB was not the first line therapy) which may also contribute to the highly variable response to ICB. In addition, the agents may be different (single or combo), and number of cycles, whether received steroid therapy, whether tumors are MSI or had deficiency in the DNA repair pathways, tumor stage, etc. however, these factors were not carefully evaluated in their analysis. Even modest difference was reported in non-small cell lung cancer, it is, however, unknown whether it was a likely “true” correlation or just co-incidence. And, the mechanistic insights are missing.

- ***To address the comment with respect to line of therapy which the reviewer raises, overall survival and time to treatment failure was stratified by line of therapy (Figure 4). Further, our multivariate model in Table 3 is adjusted for line of therapy. Steroid therapy is rarely administered, and even if given for ICB related toxicity, has not been shown to impact response to ICB. Data recently presented by our group at the 2020 GU ASCO meeting showed that the prevalence of MSI high tumors in RCC is exceedingly rare, with only one case out of 953 patients in MSK-IMPACT. Further, this case was of chromophobe histologic subtype, with no cases of MSI high tumors in clear cell RCC. We are currently examining the prevalence of DNA repair pathway deficiencies, which also appears to be rare in RCC. As pointed out in the discussion section of the manuscript regarding the effect of different features on outcome, we attempted to correct for this in our overall analysis survival (by correcting for IMDC status and line of therapy, knowing that the vast majority of therapy is given as 1st line) but agree that heterogeneity is a relevant factor.***

8. Immune deconvolution using bulk analysis is not convincing, given the heterogeneous and dynamic feature of the tumor cells and its immune microenvironment and the complex interplay between them.

- *We appreciate the limitations of utilizing bulk transcriptomic data to infer immune and stromal populations, however, we and other have validated the approaches used in this study (ssGSEA) through various orthogonal methods such as immunofluorescence and flow cytometry (Şenbabaoğlu et al, Genome Biology, 2016). Additionally, we have added IHC validation of CD31 for angiogenesis signature, and PDL1 and CD8 for the immune deconvolution component (Figure 6B).*

Reviewer #3 (Remarks to the Author):

Hakimi et al have assembled an interesting look at PBAF complex mutations first in the TCGA database and then within the MSKCC series, particularly for 189 patients with clear cell renal cell carcinoma treated with immunotherapy checkpoint inhibitors. They have connected the complicated interactions of the PBAF complex with both angiogenesis as well as the immune response.

This series of patients and their outcomes runs counter to the recently published studies from Miao et al and Braun et al, of the association of PBRM1 loss of function mutations with responses to checkpoint inhibitors. The larger database of patients in TCGA only showed the higher frequency of PBRM1, ARID2, and BRD7 mutations. The smaller set of 189 patients with ccRCC had the survival and time to treatment failure outcomes to immune checkpoint inhibitors, to evaluate the effect of PBRM1/ARID2 mutations.

The manuscript would benefit from several additions:

1. LOF mutations of PBRM1 were only 3.9% (line 73) in the MSKCC cohort, therefore potentially only 7-8 patients total. Would the authors add to the baseline characteristics Table 2 the numbers of patients who had LOF mutations in PBRM1 and ARID2, and whether they were treated with single agent IO, IO-IO combo, or other?

- *We appreciate the reviewers' comments and suggestions. We believe that the 3.9% of LOF mutations of PBRM1 that the reviewer is referring to is in the pan-cancer cohort (n=2,936), not in the RCC cohort. We have also added to Table 2 the number of patients with LOF vs. non-LOF mutations, and whether they were treated with single agent IO, IO-IO combo, or other, as per the reviewers' suggestion.*

2. The Braun et al publication showed a significantly higher proportion of patients with PBRM1 LOF mutations had objective responses to nivolumab versus the proportion in non-responders. Could the authors present the objective responses to immunotherapy treatments, for the 189 patients treated at MSKCC, and show proportions of patients who had LOF mutations in PBRM1 and ARID2 in the responders vs. non-responders?

- *As this is not a clinical trial dataset, we unfortunately do not have RECIST data for this cohort of patients and are unable to examine objective responses as was done in the aforementioned Checkmate 025 trial, for example. In lieu of RECIST data, we examined time to treatment failure.*

3. The introduction or discussion of the Braun et al manuscript (line 40 or line 220) would benefit from acknowledgement that the PBRM1 mutations were present but not sufficient for IO response (higher proportion in the responders, statistically significant but numerically similar than the non-responders).

- *This is an important point which the reviewer raises, and has been added to the discussion of the Braun et al manuscript.*

4. PBRM1 is located on chromosome 3p21. Since the authors state that PBRM1 loss seems to be more linked with dysregulated angiogenesis – what is the postulated association with VHL loss on 3p26-25? These loci are close in physical location – are they lost together? What, if any, is the association with chromosomal instability?

- *Elegant pre-clinical data support the idea that VHL (the most commonly mutated gene in clear cell RCC) and PBRM1 co-occur; although VHL is the initial driver event in the pathogenesis of clear cell RCC, genetic deletion of VHL in mice is insufficient to initiate kidney tumors [Nargund et al, Cell Rep, 2017]. This study demonstrates that after loss of VHL, loss of additional 3p21 tumor suppressor genes (PBRM1) further activates HIF1/STAT3 signaling in mouse kidney and positions mTORC1 activation as the third preferred driver event.*

5. Minor comments:

Intermittent typos should be fixed:

Line 114 ARID2 should be italicized

Line 152 univariate OR (not of) multivariate tests

Line 157 PBRM1 mutants (not mutatants)

Line 164 IFNg is or serves as a critical cytokine (not “is serves”)

- *These typos have now been fixed in the manuscript.*

REVIEWERS' COMMENTS (SECOND ROUND OF REVIEW):

Reviewer #2 (Remarks to the Author):

1. The authors should state this limitation in the manuscript.

- *This limitation is mentioned in paragraph two of the results section.*

2. The authors performed additional analysis on splice site mutations, I'm satisfied with the analysis.

3. For the study that the authors referred to (Ho et al, Urol Oncol, 2015) or other public datasets, are the authors able to find out what fraction of protein loss was caused by genomic mutations in the PBAF complex? This can also help answer the point #4.

- ***Indeed, determining the fraction of protein loss caused by genomic mutations in the PBAF complex would validate and strengthen the notion supported by IHC data that protein loss is similar to PBRM1 mutation rate in clear cell RCC. In a 2012 Nature Genetics study, Brugarolas et al. sequenced PBRM1 in 176 ccRCCs, identifying 92 somatic mutations. The authors then correlated sequencing data with results from IHC; ~90% of samples that were negative for PBRM1 by IHC had a mutation, and ~90% of the samples that were positive were wild type ($p=4 \cdot 10^{-23}$).***

5. Based on the Table 3, the clinical outcome is clearly impacted by line of therapy and drug class, the authors observed no significant difference in OS or TTF between patients with PBRM1 LOF mutations vs. wild type at pan-cancer level, were there any difference in patient's response rate or progression rate at pan-cancer level or in specific cohort(s)? Given that 51% of the ccRCC patients received ICB as first line (Table 2), any difference in the mutation frequency of PBAF complex in ccRCC patients (or other cancer types) who received ICB as 1st line of therapy vs. refractory/relapsed patients who received ICB as ≥ 2 lines?

- ***There is no significant difference in PBRM1 frequency between patients treated at 1st vs >1 st line (Fisher's exact test $p=0.56$). For LOF only mutations, $p=0.54$, and likewise for ARID2, $p>0.99$. Further, we do not have response rate or progression-free survival for the pan-cancer cohort.***

Reviewer #3 (Remarks to the Author):

The authors have addressed all of the reviewer comments, with improvements throughout from these revisions.